# Activated Carbon from Corncobs Doped with RuO$_2$ as Biobased Electrode Material

**Viola Hoffmann** [1,*], **Catalina Rodriguez Correa** [1], **Saskia Sachs** [1], **Andrea del Pilar Sandoval-Rojas** [2,3], **Mo Qiao** [4], **Avery B. Brown** [5], **Michael Zimmermann** [6], **Jenny Paola Rodriguez Estupiñan** [3], **Maria Teresa Cortes** [3], **Juan Manuel Carlos Moreno Pirajan** [3], **Maria-Magdalena Titirici** [4] **and Andrea Kruse** [1]

1 Department of Conversion Technologies of Biobased Resources, Institute of Agricultural Engineering, University of Hohenheim, Garbenstrasse 9, 70599 Stuttgart, Germany; catalina.rodriguez@uni-hohenheim.de (C.R.C.); grimmsachs@aol.com (S.S.); andrea_kruse@uni-hohenheim.de (A.K.)

2 Departamento de Química, Universidad Nacional de Colombia-Sede Bogotá, Av. Carrera 30 #45-03, Bogota 111711, Colombia; asandovalro@unal.edu.co

3 Departamento de Química, Universidad de los Andes, Carrera 1 #18A-12, Bogota 111711, Colombia; Jp.rodrigueze@uniandes.edu.co (J.P.R.E.); marcorte@uniandes.edu.co (M.T.C.); jumoreno@uniandes.edu.co (J.M.C.M.P.)

4 Department of Chemical Engineering, Imperial College London, South Kensington Campus, London SW72AZ, UK; mo.qiao@qmul.ac.uk (M.Q.); m.titirici@imperial.ac.uk (M.-M.T.)

5 Worcester Polytechnic Institute, Department of Chemical Engineering, 100 Institute Road, Goddard Hall 129, Worcester, MA 0160, USA; avebbrown@gmail.com

6 Karlsruhe Institute for Technology (KIT), Institute for Catalysis Research and Technology (IKFT), Hermann-von-Helmholtz-Platz 1, 76344 Eggenstein-Leopoldshafen, Germany; michael.zimmermann@kit.edu

* Correspondence: vio.hoffmann89@gmail.com; Tel.: +49-176-75034906

**Abstract:** Bio-based activated carbons with very high specific surface area of >3.000 m$^2$ g$^{-1}$ (based on CO$_2$ adsorption isotherms) and a high proportion of micropores (87% of total SSA) are produced by corncobs via pyrolysis and chemical activation with KOH. The activated carbon is further doped with different proportions of the highly pseudocapacitive transition metal oxide RuO$_2$ to obtain enhanced electrochemical properties and tune the materials for the application in electrochemical double-layer capacitors (EDLC) (supercapacitors). The activated carbon and composites are extensively studied regarding their physico-chemical and electrochemical properties. The results show that the composite containing 40 wt.% RuO$_2$ has an electric conductivity of 408 S m$^{-1}$ and a specific capacitance of 360 Fg$^{-1}$. SEM-EDX, XPS, and XRD analysis confirm the homogenous distribution of partly crystalline RuO$_2$ particles on the carbon surface, which leads to a biobased composite material with enhanced electrochemical properties.

**Keywords:** advanced carbon; biobased conductive carbon; energy storage

## 1. Introduction

The last two centuries were characterized by rapid industrial and technological progress based on fossil resources like oil, gas, and coal. Meanwhile, it is undisputable that this fossil resource base will not last for a further 200 years. Therefore, a lot of research has been performed and much remains in order to create a new sustainable and reliable resource base. That is exactly what the concept of bioeconomy is aiming at—the development of new materials based on renewable resources which are not only ecologically and socially acceptable, but also economically feasible [1].

One possible application for such new materials can be found in the large field of energy storage, which becomes increasingly important especially in times of renewable energy technologies which lack reliable storage systems or electric mobility. Supercapacitors or electrochemical double-layer capacitors (EDLC) are playing a key role in this context as they close the gap between batteries and conventional capacitors by providing high specific

capacities in short time. Because of the wide range of applications of these supercaps (e.g., electric or hybrid mobility, wind turbine, or photovoltaic systems), they recently gained an increased interest. A lot of research has been done in order to develop bio-based, highly porous materials which can substitute the conventional fossil-based electrode materials. As the charge-storage mechanism in EDLCs is based on electrostatic adsorption and redox reactions (=pseudo-capacitance effect) on the surface of the conductive electrodes, the viability of a new electrode material highly depends on the respective porosity or specific surface area (SSA) [2].

Carbonaceous materials with their great range of allotropes (e.g., graphite, diamond), nanostructures (graphene, nanotube, fullerene), and good physicochemical properties (chemically and thermally stable) have been well-proven in electrochemical applications regarding specific capacitance and cycling stability [3]. Nevertheless, they are facing the problem of raising cost as fossil resources will be even more scarce in future. For that reason, bio-based carbonaceous materials characterized by a lower production cost (due to use of low-value biomass) will be competitive as long as they reach the same electrochemical performance as state-of-the-art materials. Trials with activated HTC (hydrothermal carbonization) and pyrolysis carbons of different precursors have already been conducted and showed that, under certain conditions (organic electrolyte, HTC of cellulose, starch, sawdust activated at 700 and 800 °C with KOH/sample weight ratio of 4), the bio-based materials perform even better than the conventional one regarding the specific capacitance (=236 Fg$^{-1}$ at 1 mVs$^{-1}$ for bio-based electrodes) [4].

Furthermore, the electrochemical performance of activated HTC carbons containing heteroatoms (e.g., nitrogen) or metal-oxides (e.g., ruthenium oxide RuO$_2$) has been studied with the result that the specific capacitances of the functionalized or composite materials reached values up to 300 Fg$^{-1}$ and 333 Fg$^{-1}$, respectively. It is assumed that the combination of capacitance and enhanced pseudocapacitance leads to this superior capacitance values [4–6].

Nevertheless, it still remains unclear, which process leads to the most viable material for application in EDLCs. As not only the SSA, but also the pore size distribution (PSD) and the surface functionality play a major role in this context, different nanocasting methods of the carbonaceous material have been tested (e.g., different activation methods; HTC coating techniques with Mn, Fe, and Co; HTC of silicon nanoparticles and glucose at 180 °C) [7].

In addition, different biomasses have been used as precursor and the activation temperatures have been varied in order to understand which combination leads to which physico-chemical properties of the activated carbon [8,9].

Another application for the mentioned bio-based materials would be in the field of energy generation, or more specifically as electrodes in direct carbon fuel cells (DCFC). As the required physico-chemical and electrochemical characteristics for electrodes in EDLCs or DCFCs are pretty similar, it makes sense to not only focus on the field of energy storage, but also on the no less up-to-date field of energy generation.

This work focuses on the production of bio-based materials with similar or even better electrochemical properties when compared with fossil-based materials for electrodes energy storage or conversion technologies. Corncob was used as precursor for the production of activated carbon, which was then modified by adding the metal-oxide RuO$_2$ and additives. The produced materials were characterized with respect to their physico-chemical and electrochemical properties with the aim to draw conclusions about correlations between the physico-chemical properties and the electrochemical performance of these materials.

## 2. Materials and Methods

### 2.1. Materials

The materials for the production of the active electrode material were partly produced at the University of Hohenheim and partly commercialized by Sigma-Aldrich Chemie GmbH (Taufkirchen, Germany) and VWR International GmbH (Darmstadt, Germany). The activated carbon was produced by pyrolysis of corncobs (under nitrogen gas and a constant

heating rate of 10 K min$^{-1}$) and a following KOH activation (ratio 4:1) of the pyrolysis char at 800 °C (elemental composition see Table 1). The obtained activated carbon was then used to produce composites with different contents of RuO$_2$. The composites were produced by a sol-gel process using the produced activated carbon, RuCl$_3$, and NaOH. Composites with different contents of RuO$_2$ were obtained and analysed with respect to their physico-chemical properties and their conductivity before they were applied to a current collector (primered Al-foil) in order to determine their capacitive behavior by using cyclic voltammetry.

**Table 1.** Elemental composition of the corncobs and the activated carbon used as precursor for the production of the electrode material. The elementary composition is given on a dry and ash free (DAF) basis. The ash content was calculated by difference.

|  | Corncobs | Activated Carbon (AC-600-800) |
|---|---|---|
| C$_{DAF}$ (wt.%) | 47.9 | 95.9 |
| H$_{DAF}$ (wt.%) | 6.5 | 0.1 |
| N$_{DAF}$ (wt.%) | 2.9 | 0.3 |
| S$_{DAF}$ (wt.%) | 0.1 | 0.0 |
| O$_{DAF}$ (wt.%) | 42.6 | 3.7 |

### 2.1.1. Activated Carbon (AC)

The activated carbon was produced using corncobs collected from the agricultural experiment station Heidfeldhof of the University of Hohenheim. The corncobs were divided in three prior to the pyrolysis. Approximately 100 g of corncobs were heated up to 600 °C in a cylindrical quartz glass tube reactor heated externally with a split tube furnace ROK 50/250/11 (Thermo Fisher Scientific, Frankfurt, Germany) with a constant heating rate of 10 K min$^{-1}$ under nitrogen gas. When the final temperature was reached, the samples were left to react for 2 h under a constant nitrogen flow of 3 L min$^{-1}$. After the reaction time was over, the chars were cooled down naturally to room temperature, milled using a Pulverisette 6 ball mill (Fritsch GmbH, Fellbach, Germany), and sieved to a particle size smaller than 500 μm.

To increase the surface area of the pyrolysis chars (P600), a chemical activation process was conducted using KOH as activating agent. For this, a mixture with a char to KOH mass ratio of 1:4 was produced in nickel crucibles. The crucibles were placed inside a batch stainless steel reactor, which in turn was placed inside a muffle furnace that had been previously heated to 600 °C. The samples were heated up to this temperature in approximately 20 min and were left to react for 2 h. Subsequently, the reactor was removed from the furnace and was left to cool down under a constant N$_2$ flow of 20 L min$^{-1}$ to avoid spontaneous combustion. Then, the samples were washed with boiling water until the leachate reached a constant electrical conductivity. Afterwards, the sample was washed with approximately 250 mL of an aqueous HCl solution (2 M) to ensure the removal of all the potassium compounds. Finally, the samples were washed with hot deionized water until the electric conductivity was lower than 20 μS cm$^{-1}$ and the pH became nearly neutral. The samples were dried at 105 °C for at least 16 h and stored for further processing and characterization. The activation yield was approximately 50%, whereas the process yield (pre-carbonization and activation) was about 15%.

The elemental analysis of the corncobs and of the activated carbon were determined with a Euro EA—CHNSO Elemental Analyser (Hekatech GmbH, Wegberg, Germany). The ash content was determined by difference according to Equation (1) (in wt.%).

$$\% \; Ash = 100 - (\% \; C + \% \; H + \% \; N + \% \; S + \% \; O)\% \qquad (1)$$

### 2.1.2. Composite Production

The composite production was carried out on the basis of a sol-gel method described elsewhere [6]. In order to obtain different contents of $RuO_2$, stoichiometric calculations were applied to determine the respective amounts of $RuCl_3$ which are necessary to reach 10 wt.%, 20 wt.% and 40 wt.% of $RuO_2$ in the composite. A 0.1 mol $L^{-1}$ $RuCl_3$-solution was prepared with the respective amount of $RuCl_3$ and distilled water. Meanwhile, a homogenous dispersion was produced by mixing the respective amount of AC (AC-600-800) with the same amount of distilled water. After stirring for half an hour, the $RuCl_3$-solution and the AC dispersion were mixed and stirred for another half an hour. Then, a 0.1 mol $L^{-1}$ NaOH-solution was added with a pH-titrator till the pH was 7. The obtained mixture was stirred for six hours before vacuum filtering it to obtain the precipitate (AC-$RuO_2$-x; x stands for the respective percentage of $RuO_2$ contained in the composite). In order to clean the composite material from residual NaCl, a Soxhlet filtration with distilled water was conducted overnight. The Ru-content of the filtration water was determined by ICP-OES measurements (Agilent 715, München, Germany). In order to be sure that all NaCl was washed out, the composite was washed after Soxhlet-filtration several times with distilled water, and conductivity measurements of the filtration water were performed with a pH meter till the pH value of the filtration water reached the pH value of distilled water.

### 2.1.3. Electrode Preparation

In order to conduct electrochemical measurements, the working electrodes for CV measurements were produced in two different ways, depending on the electrolyte used during CV (6 M KOH or 0.5 M $H_2SO_4$). The working electrodes for CV analysis in 0.5 M $H_2SO_4$ were prepared by dry-mixing the respective active material (AC or AC-$RuO_2$ composites) with 5 wt.% binder (PTFE by VWR International GmbH, Darmstadt, Germany) and 15 wt.% conductive additive (carbon black, 99.9% by VWR International GmbH, Darmstadt, Germany) with a cryomill. The produced powders were pressed onto the current collector (20 μm thick) with a hydraulic press (with 40 t or 350 kg $cm^{-2}$). A primed current collector was used in order to achieve a better adhesion of the active material layer (primer material: thin layer of acetylene black (AB) (1 μm)). The content and sort of binder was chosen considering the work of Zhu et al. and Abbas et al. [10], who came to the conclusion that PTFE is more suitable for this kind of application than PVDF (as the latter seems to lead to a lower accessible pore volume [11]) and that the ideal binder content in order to reach the best capacitive behavior of the electrode material is 10 wt.% for PTFE [11].

The working electrode for CV analysis in 6 M KOH electrolyte was prepared by coating the active material on nickel foam as current collector. Meanwhile, 10 wt.% of PTFE were used as binder and 10 wt.% Vulcan XC72 as conductive additive.

Besides the AC-$RuO_2$-composite materials, the active materials included one commercial AC (Norrit) with a specific surface area (SSA) of 875 $cm^2$ $g^{-1}$, as well as the pure AC from corncobs produced at the University of Hohenheim (UHOH) with a SSA of 2333 $cm^2$ $g^{-1}$ (AC-600-800). Furthermore, a piece of primed current collector (AB) without any active material was tested for reference purposes. The final electrodes had an area of 1 $cm^2$ for the Al-foil current collector and an area of 1.54 $cm^2$ for the nickel foam current collector.

### *2.2. Physico-Chemical Characterization*

### 2.2.1. Specific Surface Area (SSA) and Pore Size Distribution (PSD)

The specific surface area (SSA) and pore size distribution (PSD) was determined using gas adsorption measurements of the different materials by using a NOVA 4000e (3P Instruments, Odelzhausen, Germany). The $N_2$ and $CO_2$ adsorption isotherms were measured at $-196$ °C and 0 °C, respectively. The BET surface area was calculated from the $N_2$ and $CO_2$ isotherms following the model proposed by Bruanauer–Emmett–Teller



model [12] and the modifications proposed by Rouquerol et al. [13]. The relative pressure range during the measurement was between 0.05 to 0.35 p/p°. The microporosity and mesoporosity were estimated by applying the t-plot and Barrett–Joyner–Halenda (BJH) models [14] to the $N_2$ isotherm, respectively. The pore size distribution was analyzed by applying the non-local density functional (NLDFT) theory [15,16].

### 2.2.2. Immersion Calorimetry (IC)

Immersion calorimetric experiments were performed using a Calvet type heat conduction calorimeter, the solvent used were water and benzene in order to characterize accessible surface area and the hydrophobic factor. The calorimetric measurements were performed at 19 °C under the following experimental conditions. The baseline signal was recorded 1 h prior to the immersion until it reached stability (±2.0 microvolts), and the baseline should be reached once the immersion and calibration process occur. Constant calorimeter is approximately 700 W/V for an electrical work that dissipated about 6.0 J, with a sensitivity of $50 \times 10^{-3}$ V/W and a deviation less than 2%.

To perform the calorimetric experiments, 0.050 g activated carbon samples were placed in a glass bulb. The samples should be outgassed at 160 °C for 8 h under vacuum. After this process, it is guaranteed that the surface of the solid is clean and free to interact with the molecules of the solvents. To determine the immersion enthalpies, the previously degassed glass bulbs were attached to the calorimetric cell with 8 mL of the desired solvent and the calorimeter was assembled. Once the baseline was reached, i.e., thermal equilibrium has been achieved, the cell is immersed in the solvent. The breaking of this bulb produced a calorimetric effect of about 5 μV which did not affect the overall value within the enthalpy. The resulting immersion thermal effect was recorded by the sensors (thermopiles) until a stable baseline was obtained. Recordings were then continued for an additional time period of 20 min after immersion, followed by electrical calibration of the calorimeter. Experiments were repeated about four times for each solvent listed above.

Water is used as probe molecule of the changes in the surface chemistry of porous solids, due to the treatments to which the original solid is subjected, so the enthalpic contributions of the higher energy sites on the surface will give rise to specific interactions such as hydrogen bonds. On the other hand, benzene is considered an organic molecule with little possibility of producing specific interactions, as well as hydrogen bonds and polar junctions. Instead, it produces low energetic donor-acceptor interactions with delocalized π electrons of the graphene layers of solids like activated carbons (π stacking interactions), and for this reason it can be used as a non-polar reference. In the absence of specific interactions between the immersion liquid and the solid surface, immersion calorimetry is a valuable method to assess the accessible area since the immersion enthalpy is proportional to the surface area evaluated by $N_2$ isotherms. Another parameter that can be evaluated by immersion microcalorimetry is a change in the hydrophilicity of the surface by the relationship between the enthalpy of water immersion (polar) and a non-polar solvent, in this case benzene [17–19].

### 2.2.3. Zeta-Potential

The values of zeta potential for the different samples were obtained by measurements with a Zeta Nano ZS (Malvern Instrument, Malvern, UK) equipped with a high concentration cell. For measuring the samples, a suspension was first prepared by mixing the powder with deionized water. The first zeta potential was measured at the current pH value. Then, NaOH (0.1 M) was added till the pH reached values >10. The suspension was then titrated with HCL (0.1 M) to reach the pH values 9, 8, 7, 6, 5, 4, 3 and 2. At each of these pH values, the zeta potential was measured three times.

### 2.2.4. X-ray Diffraction (XRD)

X-ray Diffraction Measurements were carried out with a X'Pert MPD (PANalytical GmbH, Kassel, Germany) with a Bragg-Brentano, Cu Kα source, Ge-(111)-Johannson single

crystal monochromator and 1D Silicon Strip Detector. The measurements were conducted as following: a two-side adhesive tape was fixed on a piece of glass ($2 \times 2$ cm$^2$) and the sample powder was spread into the tape. A flat layer of the powder was formed by pressing it. The piece of glass was put on the sample holder and scanned with X-ray by an angle of two theta 20–60 since this is the range of almost all important peaks of the measured materials.

### 2.2.5. X-ray Photoelectron Spectroscopy (XPS)

The XPS spectra were measured on a Thermo Fisher (Frankfurt, Germany) K-Alpha+ X-ray photoelectron spectrometer operating at $2 \times 10^{-9}$ mbar base pressure. The system uses a monochromated, microfocused Al K$\alpha$ X-ray source (h$\nu$ = 1486.6 eV) and a 180° double focusing hemispherical analyser with a 2D detector. The X-ray spot size is 400 µm$^2$ provided by X-ray sources operated at 6 mA emission current and 12 kV anode bias. Advantage Data System software was applied to record the XPS spectra through the whole measurement. Advantage software was used for the quantitative analysis of spectra.

### 2.2.6. Scanning Electron Micrsocopy (SEM)

SEM analyses were performed with a LEO Gemini 982 from Zeiss (Oberkochen, Germany) equipped with an annular high brightness in-lens-SE detector for high resolution and true surface imaging. A laterally mounted SE detector (Everhart-Thornley-type) provides topographical contrast (sensitive for SE+BSE). The beam accelerating voltage was 10 kV. The determination of the chemical composition was performed using an Oxford INCA Penta FETx3 EDX system (Abingdon, UK).

### *2.3. Electrochemical Properties*
### 2.3.1. Electric Conductivity (EC) Measurements

Conductivity measurements were conducted by using a self-constructed device based on a procedure described elsewhere [20–22]. The device consists of two parts: the bottom part is made of a brass bracket with a fixed glass cylinder and the upper part is a movable brass-stamp on which different pressures can be exactly applied using a material testing system of Instron (Instron GmbH, Darmstadt, Germany) with an integrated altimeter for determining the height of the respective sample after applying different pressures. The masses of AC and nanocomposites used, after oven-drying at 105 °C overnight, were around 0.09 g. Each sample was poured into the hollow thick-walled glass cylinder (inner diameter of 0.1 cm), and compressed in air between the two close-fitting brass plungers forming the electrodes. The resistance was measured with a multimeter. The resistance of each sample was measured under different pressures induced by weights (2 kg, 5 kg, 10 kg). Before measuring the resistance, the weights were applied to the upper plunger for 5 min in order to get a regular contact surface between the sample and the stamp. Each sample was measured three times. The conductivity was calculated by using the following formula

$$\sigma = \frac{h}{R \cdot A} \tag{2}$$

with

$\sigma$ = electrical conductivity [S m$^{-1}$]
$h$ = height of the sample [m]
$R$ = electrical resistance [$\Omega$]
$A$ = area of the sample [m$^2$]

### 2.3.2. Cyclic Voltammetry (CV)

CV measurements were performed in a three-electrode cell [23] by using a PG-STAT302N potentiostat/galvanostat (Autolab, Deutsche METROHM GmbH & Co. KG, Filderstadt, Germany). The working electrode was fixed to a copper cable with silver conductive paint and epoxy resin (LOCTITE 2052208, Henkel, Düsseldorf, Germany) was

used to completely cover the non-active parts of the Al-foil. The reference electrode was an Ag/AgCl Reference Electrode (3 M NaCl, BASi RE-5B, MF-2030) and the counter electrode was a Pt wire. For the measurements in KOH electrolyte an Hg/HgO reference electrode was used.

## 3. Results and Discussion

### 3.1. Physico-Chemical Properties of the Activated Carbon

Table 1 shows the chemical composition of the corncob and of the activated carbon. The different thermal treatments conducted with the corncob to obtain the activated carbon led to a drastic increase of the carbon content and a decrease of the other elements as well as ash. The reduction of the ash content is a consequence of reactions between the inorganic components in the biomass and the KOH leading to soluble compounds [9].

### 3.1.1. SSA and PSD

The activated carbon produced from corncob developed an extremely large surface area composed mainly of micropores (87% of the total surface area is microporous; see Table 2). The surface areas calculated from the $CO_2$ isotherm were larger than those calculated from the $N_2$ isotherm, which indicates a restricted diffusion of $N_2$ compared to $CO_2$ due to the narrow microporosity of the carbon [24]. The addition of additives (binder) to the carbon led to a large decrease of the surface area, due to pore blockage. However, the isotherm shape and the values calculated from the t-plot indicate that the doped carbons remained a microporous material (Figure 1). The surface area calculated from the $CO_2$ isotherm of the carbon with binder was similar to that calculated from the $N_2$ isotherm and the microporous volume did not decrease considerably (Figure 1A,B). The addition of $RuO_2$ on the other hand resulted in a dramatical decrease of both the total and microporous surface areas. The size of the $RuO_2$ particles ranges between 15 and 45 nm [25], which is large enough to fit inside mesopores and block the micropore access. This is described by the pore size distribution graph in Figure 1C, where the number of pores in the lower size range decreases dramatically after the addition of the $RuO_2$.

**Table 2.** BET SSA of the activated carbon AC-600-800 and the composites AC-$RuO_2$-10 and AC-$RuO_2$-40.

| Sample | $S_{BET},N_2$ $(m^2/g)$ | $S_{BET},CO_2$ $(m^2/g)$ | $S_{micro}$ $(cm^2/g)$ | $V_{micro}$ $(cm^3/g)$ |
|---|---|---|---|---|
| AC-600-800 | 2334 | 3145 | 2037 | 0.89 |
| AC-600-800-binder | 1759 | 1452 | 1516 | 0.67 |
| AC-$RuO_2$-10 | 344 | 384 | 248 | 0.11 |
| AC-$RuO_2$-40 | 317 | 299 | 264 | 0.11 |

### 3.1.2. Immersion Calorimetry

Immersion enthalpy ($\Delta H_{Imm}$) is a useful thermodynamic parameter that can characterize the textural and chemical properties of a porous solid. Immersion calorimetry allows obtaining a first approximation of the surface chemistry of the solid, e.g., in terms of its affinity to polar or non-polar solvent [17–19]. The started activated carbon as the ruthenium composites were characterized by immersion calorimetry in water and benzene (Table 3).

**Table 3.** Immersion enthalpies in benzene and water and hydrophilic factor of each sample.

| Sample | $\Delta H_{Imm}C_6H_6$ [a] $(J/g)$ | $\Delta H_{Inm}H_2O$ (b) (J/g) | Hydrophilic Factor $= \frac{(b)}{(a)}$ |
|---|---|---|---|
| AC-600-800 | −218.0 | −173.1 | 0.79 |
| AC-$RuO_2$-10 | −71.71 | −132.0 | 1.84 |
| AC-$RuO_2$-40 | −204.4 | −21.24 | 0.10 |

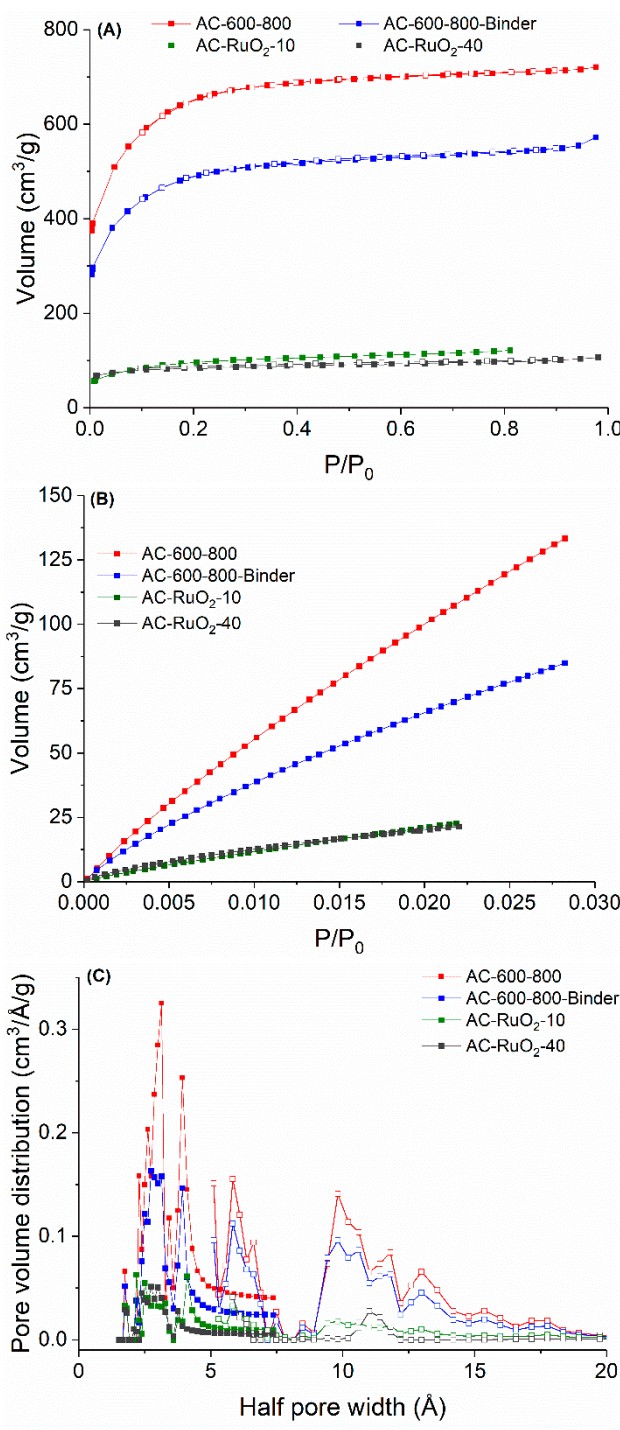

**Figure 1.** (**A**) $N_2$ adsorption (closed symbol) and desorption (open symbols) isotherms; (**B**) $CO_2$ adsorption isotherms; (**C**) pore size distribution calculated from the $CO_2$ (closed symbols) and $N_2$ (open symbols) isotherms.

Figure 2 shows the calorimetric potentiograms measured for AC-600-800 and ruthenium composites in water. The immersion enthalpies calculated were between −21.24 and −218 J g$^{-1}$ for the different ACs and probe molecules. The large differences in benzene experiments are a consequence of diffusional restrictions within the solid, which were more notable for the solids with lower surface areas (AC-RuO$_2$-10 and AC-RuO$_2$-40).

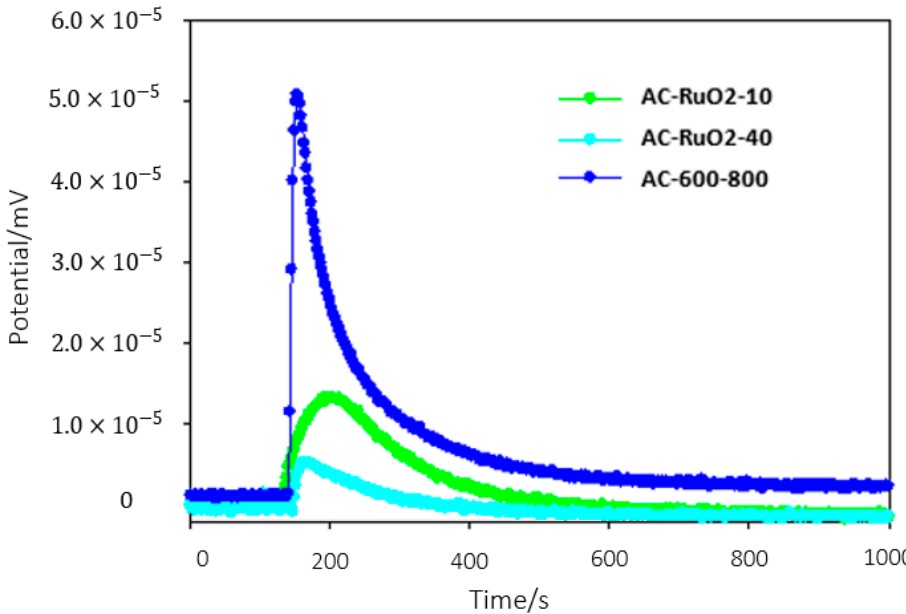

**Figure 2.** Potentiogram obtained for the immersion of solids in water.

The sample AC-RuO$_2$-10 with the lowest enthalpy in benzene ($-71.71$ J g$^{-1}$) demonstrated a restriction to the benzene molecules to access the internal porous structure. This could be related to the SEM images in which tendency of agglomeration was observed. Moreover, it has been argued that the decrease in the immersion enthalpies of solids with similar textural parameters (is this the case of AC-RuO$_2$-10 and AC-RuO$_2$-40) could be an artefact of the molecular packing within the pores, and not an indication of molecular sieve exclusion [17].

On the other hand, the immersion calorimetry in water showed different behavior for every sample. The start solid had the highest thermal effect (173.1 J g$^{-1}$) due to the free polar sites on the surface, like oxygen surface groups (carboxylic acids, lactones, and phenols), which are located at the edges of the pore openings or in defects of the graphene layers in the carbon structure. These groups could lead to specific interactions with water molecules. Therefore, the released energy can be related to the establishment of these interactions, such as hydrogen bonds [26–28]. The immersion enthalpies in water by AC-RuO$_2$-10 and AC-RuO$_2$-40 samples are lower, and this could be related to the change in surface charge due to aa decrease in the acidity of the surface functional groups with crystallization of the ruthenium oxide, due to the thermal treatment required to degas the sample prior to the immersion experiments [29]. Hydrophilic factor, obtained by the ratio of the enthalpy of immersion in water to that of benzene evidenced that the sample AC-Ru$_2$O-40 was more hydrophobic compared with Ac-Ru$_2$O-10, which is in accordance with the above discussed.

### 3.1.3. Zeta Potential

Zeta potential values at different pH-values in Figure 3 show that the composite containing 40 wt.% RuO$_2$ has a higher stability at lower pH-values than the composite with 10 wt.% RuO$_2$. Nevertheless, both composites show good stability when pH is 9 or higher. At pH 2.5–3.5, a tendency towards rapid coagulation or flocculation can be observed for AC-RuO$_2$-40, whereas AC-RuO$_2$-10 is more stable at these conditions. Overall, the RuO$_2$-impregnated samples are more electrostatically active compared to the activated carbon. At low pH values the RuO$_2$-impregnated samples have higher Zeta potentials than the activated carbon, and this promotes the agglomeration of particles. As the pH of the solution rises, the particles of all three samples begin to lose stability. The higher electrostatic activity of the AC-RuO$_2$ samples causes them to lose stability at a faster rate.

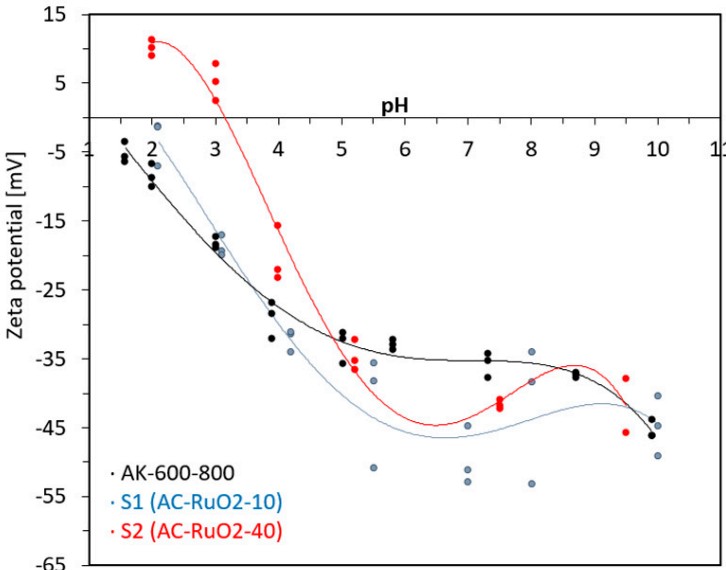

**Figure 3.** Zeta potential values at different pH-values for AC (AC-600-800), S1 (AC-RuO$_2$-10) and S2 (AC-RuO$_2$-40) (IFKB, University of Stuttgart).

### 3.1.4. Composition

EDX measurements show that the elemental compositions of the samples agree well with the theoretical, calculated expectations (see Figure 4). The EDX spectra show that for the composites with a calculated content of 10 wt.% and 40 wt.% of RuO$_2$, respectively, the contents of RuO$_2$ are even higher than the expected values. This can be explained by

(1) The roughness of the sample surface, since the evaluated L$\alpha$ spectral line of Ru has the highest energy (25,586 keV) and is overestimated in shadowed areas towards the K$\alpha$ lines of C and O.

(2) The superficial coverage of the carbon particles, since the depth of X-ray generation is conceivably not as big as the particles are. This means that the inner part of the particles is neglected by the measurement.

(3) The inhomogeneity of the sample leads to statistical errors, since only a small area of approximately 1 mm$^2$ was analyzed.

The peak for Si, which can be seen in Figure 4, shows mineral residues from the precursor biomass and is without any relevance here.

Backscattered electron (BSE) pictures of the analyzed areas of each sample show a clear difference of RuO$_2$-content between the sample with the highest (40 wt.%) and the sample with the lowest (10 wt.%) metal-oxide content (see Figure 4). The higher content of Ru results in an increased amount of bright areas due to the high density of Ru (see Figure 5 below).

The assumption that nearly all of the Ru that was added in the form of RuCl$_3$ was converted into RuO$_2$ and bound to the surface of the AC was confirmed by ICP-OES measurements, which showed that nearly no ruthenium is left in the filtration water after Soxhlet-filtration of the precipitate.

Furthermore, the SEM images show that there is a tendency towards the agglomeration of RuO$_2$ particles. RuO$_2$ appears as spherical particles with an estimated size of 10–20 nm, which are agglomerated and sometimes even lead to clogging of macropores. This finding confirms the conclusion which was drawn based on the significant decrease in SSA between untreated AC-600-800 and the composite AC-RuO$_2$-40 (see Section 3.1.1).

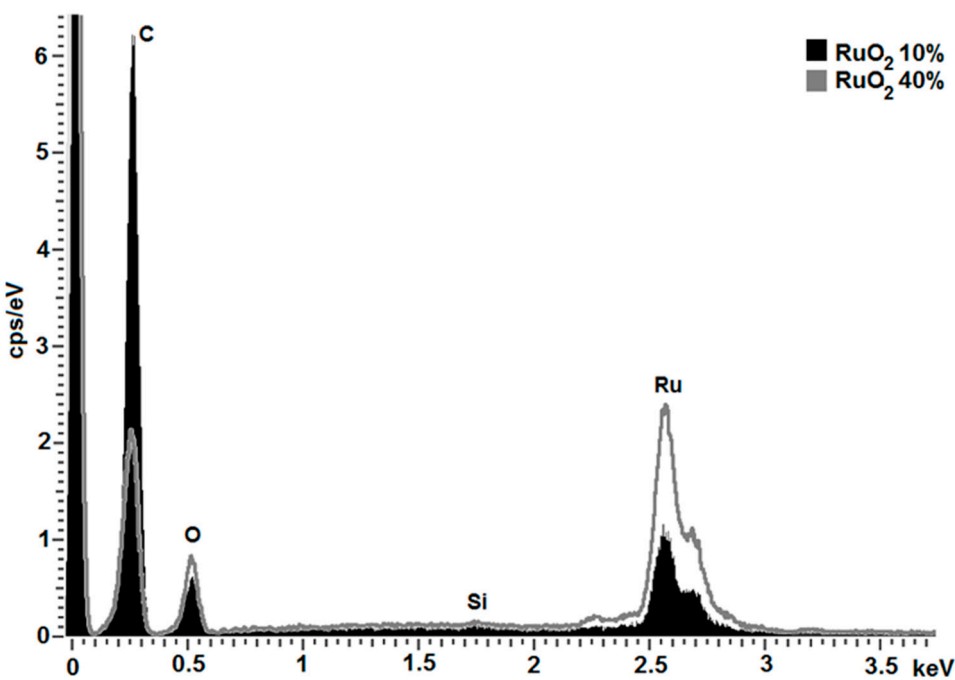

**Figure 4.** EDX spectrum of samples with $RuO_2$ with characteristic peaks for Ru, O and C.

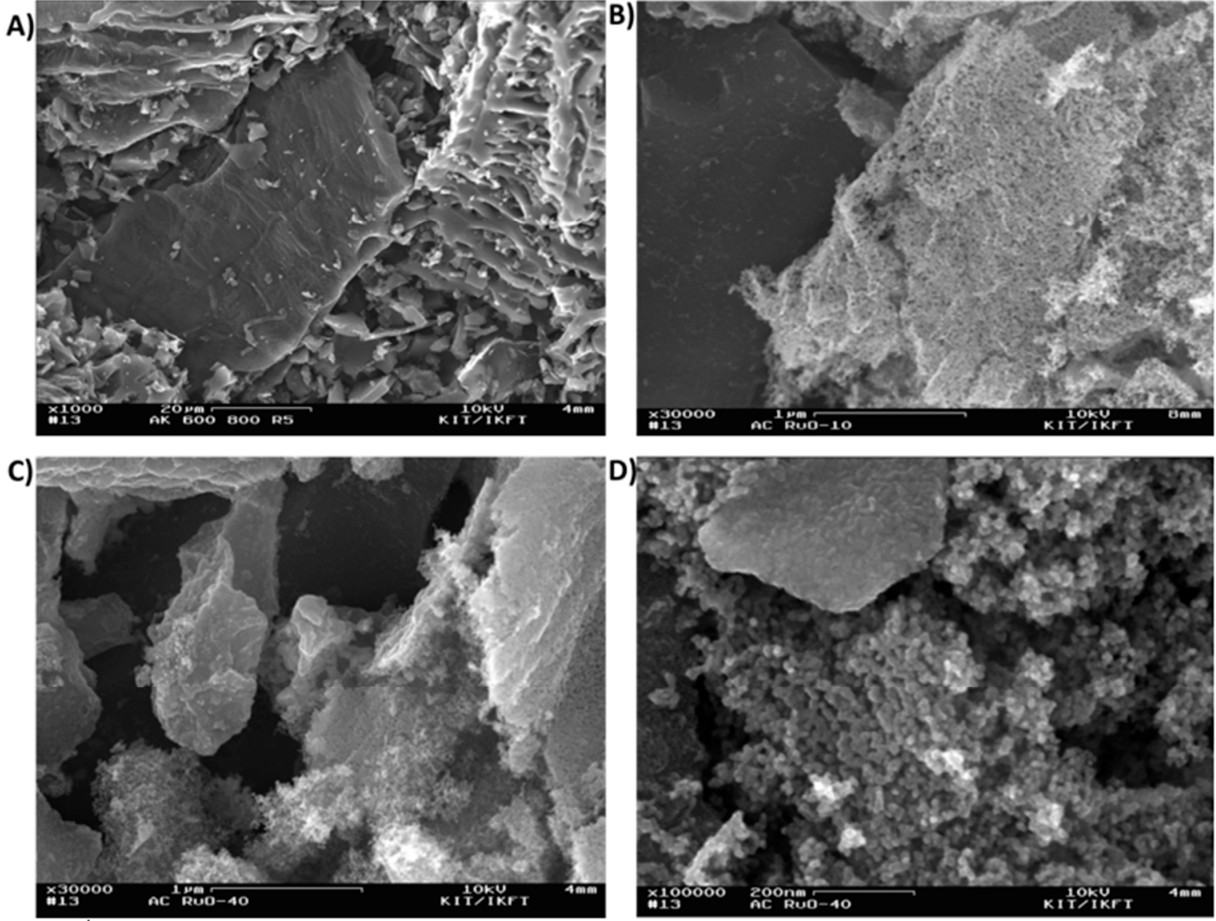

**Figure 5.** SEM pictures (Inlens SE-detector) of (**A**) AC-600-800, 1000×, (**B**) AC-$RuO_2$-10, 30,000×, (**C**) AC-$RuO_2$-40, 30,000× and (**D**) AC-$RuO_2$-40, 100,000×.

### 3.1.5. X-ray Diffraction (XRD)

Figure 6 shows the x-ray diffraction patterns of the two samples. Sample AC-RuO$_2$-10 (Figure 6A) possesses a large peak consistent with carbon. The width of this peak results from contribution of the diffraction peaks of RuO$_2$ impregnated on the surface of the carbon material. The peak position at 26° is consistent with the observations in studies of coal and biochar [30,31] and can be attributed to the spacing of aromatic ring layers and the presence of some graphite like structures. Sample AC-RuO$_2$-40 (Figure 6B) possesses several sharp diffraction peaks consistent with the presence of RuO$_2$. These patterns and locations are consistent with previous reports of Sugimoto et al., Zheng and Jow, and Foelske et al. [32–34] for hydrous RuO$_2$ and papers by Hilgendorff et al., Eblagon et al., and Gonzalez-Huerta et al. [35–37] for ruthenium black. The number of peaks suggests that there are several ruthenium phases being formed which are partly overlapping, but the additional peak at 38° can be assigned to the crystal plane (200) of RuO$_2$ based on the standard 32.1°and 37.2° (JCPDS#50-1428), supporting the assumption that crystalline RuO$_2$ was formed [38]. However, the 38° peak could also correspond to (100) crystal plane of metallic Ru (JCPDS#06-0663). Moreover, also present in the XRD pattern of AC-RuO$_2$-40 is the characteristic peak of carbon. The relative intensity of the peaks between AC-RuO$_2$-10 and AC-RuO$_2$-40 can be the result of a transition of hydrous to crystalline RuO$_2$. However, this is difficult to quantify due to the higher loading of RuO$_2$ in the second experiment.

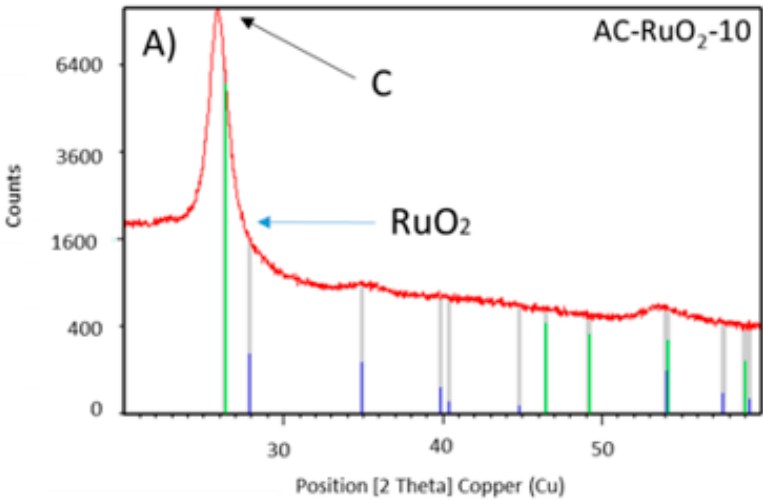

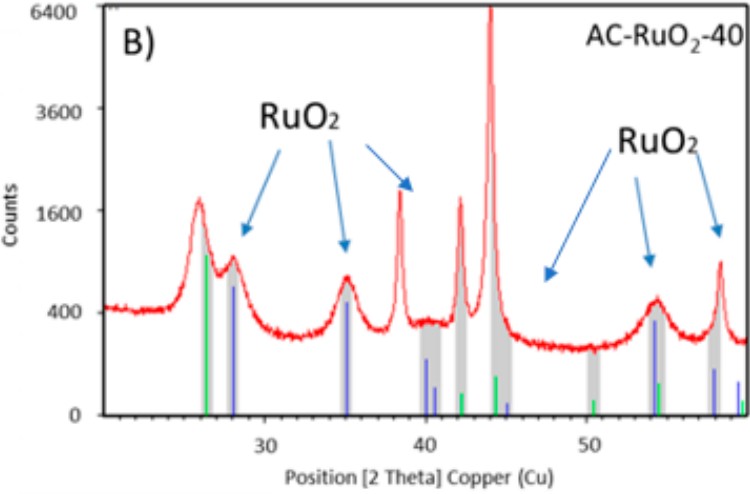

**Figure 6.** X-ray diffraction patterns of (**A**) AC-RuO$_2$-10 and (**B**) AC-RuO$_2$-40.

### 3.1.6. XPS

As shown in Table 4, the XPS survey spectra indicated that AC-RuO$_2$-10 and Ac-RuO$_2$-40 include around 4–5 at% of Ru, and a high O/C ratio compared with AC-600-800, which demonstrated the extra O introduced by RuO$_2$ in the composites. The high-resolution of Ru spectra is shown in Figure 7. For AC-RuO$_2$-10, the binding energies of Ru3d$_{5/2}$ and Ru3d$_{3/2}$ are 280.75 eV and 284.62 eV, respectively. The satellite peaks of Ru 3d$_{5/2}$ present at 282.4 eV and 287.35 eV are due to oxidation states higher than Ru$^{4+}$ (such as RuO$_3$), and the final state screen effect, respectively. The binding energy difference between Ru 3d$_{5/2}$ and Ru3d$_{3/2}$ is 3.87 eV, which is slightly lower than the reported RuO$_2$ standard. This is most probably due to the influence of the C1s peak at ca. 284.6 eV, which is attributed to the carbon matrix. Considering the C1s peak at ca. 284.6 eV is completely merged in the Rud$_{3/2}$ peak, the ratio of C1s detected is very limited, which indicated a high ratio of RuO$_2$ covering on the carbon matrix surface. As shown in Figure 8, AC-RuO$_2$-40 demonstrated a very similar configuration composition compared with AC-RuO$_2$-10. However, the Ru 3d$_{5/2}$ peak at 280.80 and Ru3d$_{3/2}$ peak at 284.88 have a binding energy difference of 4.08, which is very similar with the standard RuO$_2$ reported. This indicated a lower influence of the C1s peak and proved that the Ac-RuO$_2$-40 has a higher cover rate of RuO$_2$ on the carbon surface compared with the Ac-RuO$_2$-10, most probably due to the increased amount of Ru precursor in the synthesis process [39,40]. This is consistent with the XRD and EDS analysis above.

**Table 4.** XPS results of different samples AC-600-800, AC-RuO$_2$-10, and AC-RuO$_2$-40.

|  | C (at%) | O (at%) | Ru (at%) |
|---|---|---|---|
| AC-600-800 | 93.2 | 6.8 | 0 |
| AC-RuO$_2$-10 | 74.4 | 20.6 | 5.0 |
| AC-RuO$_2$-40 | 80.6 | 15.3 | 4.1 |

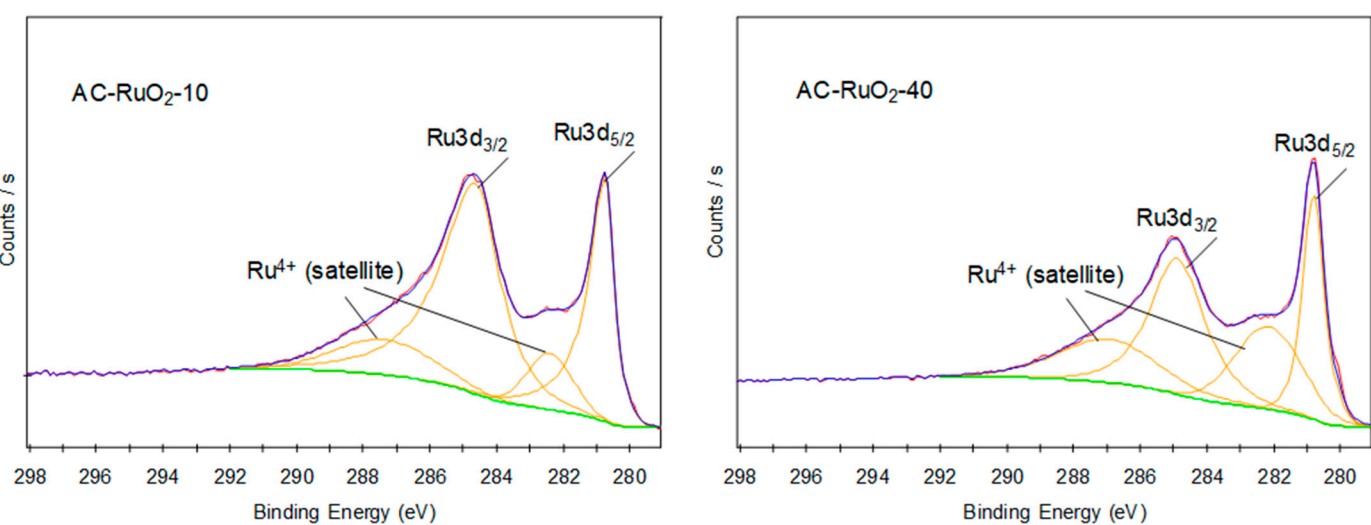

**Figure 7.** High-resolution C1s/Ru3d spectra of AC-RuO$_2$-10, and AC-RuO$_2$-40, respectively.

### *3.2. Electrochemical Performance*

### 3.2.1. Electric Conductivity (EC)

The EC measurements show that there is not only a strong, almost linear correlation between applied pressure and electric conductivity (see Figure 9), but also an evident correlation between electrical conductivity and the composition or physico-chemical properties of the measured material, respectively. For all measured samples, as expected, the conductivity values were higher when the applied pressure was increased due to an increased

particle contact area [41]. Furthermore, by comparing the three biochars P600, P800, and P900 (prepared by pyrolysis of corncobs at the respective temperatures for 2 h), it is shown that the higher the carbonization temperature, the higher the EC due to the higher level of aromatization, as discussed elsewhere [20,42]. Higher specific surface areas also lead to higher EC (AC-600-800 compared to P600) due to better particle contact [20]. As expected, adding highly conductive $RuO_2$ leads to the highest EC values of up to 408 S m$^{-1}$.

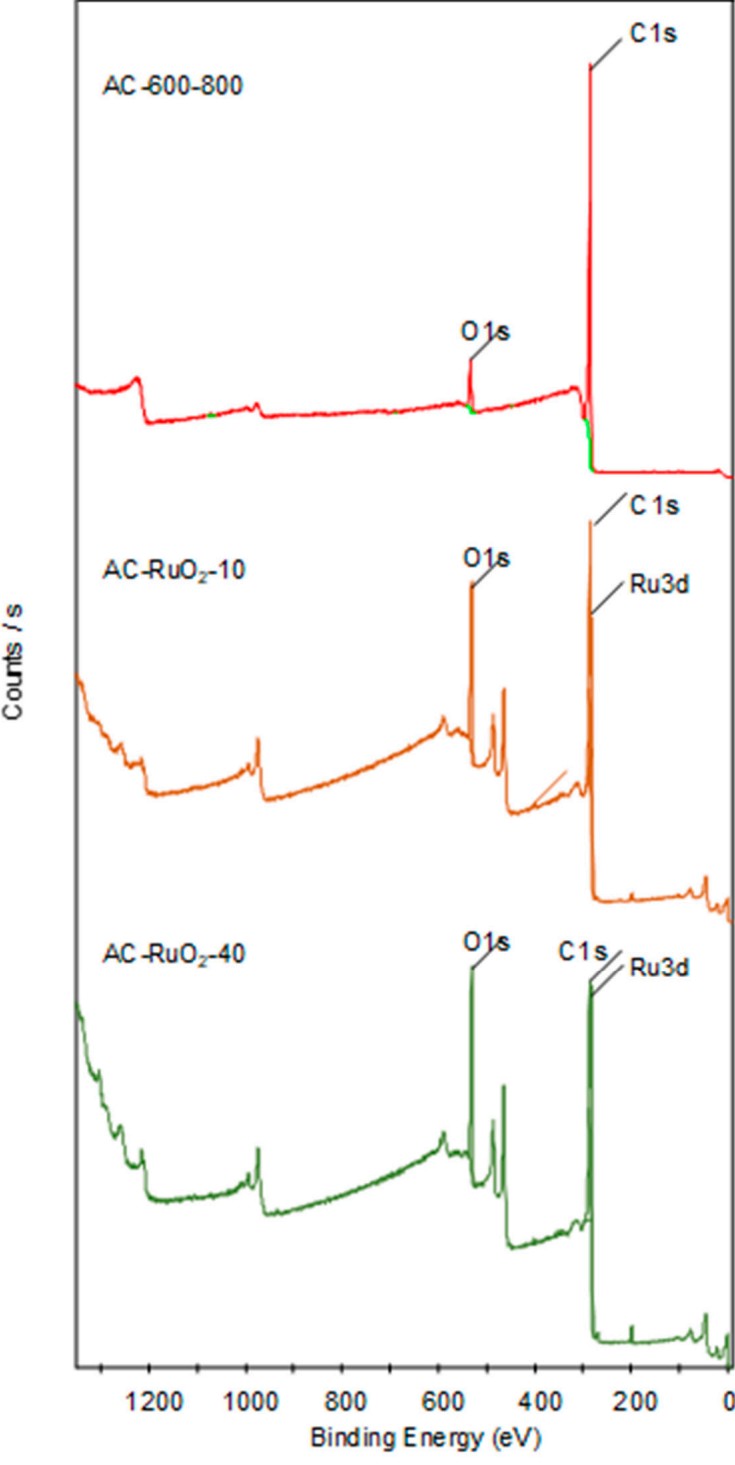

**Figure 8.** XPS survey spectra of sample AC-600-800, AC-RuO₂-10, and AC-RuO₂-40, respectively.

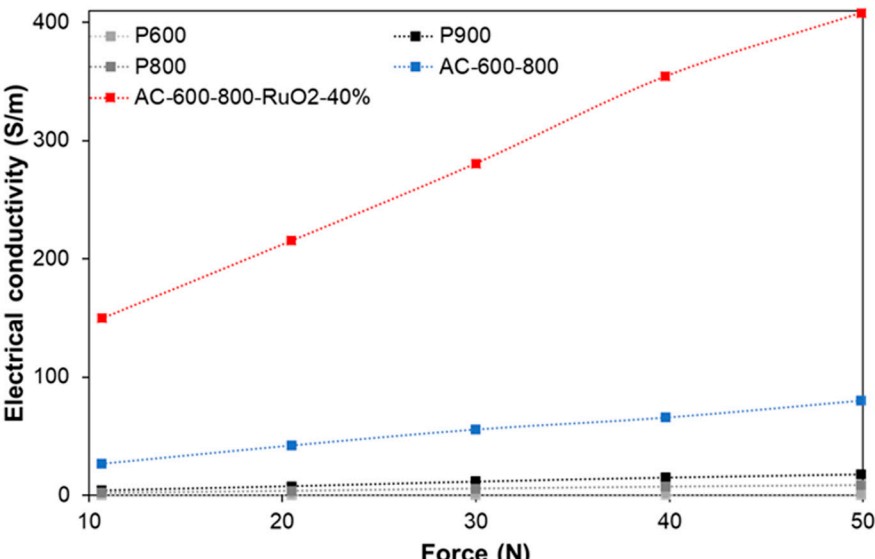

**Figure 9.** Electrical conductivity in S m$^{-1}$ vs. pressure in N for pyrochars (P600, P800, P900), AC-600-800 and the composite AC-600-800-RuO$_2$-40.

### 3.2.2. Capacitive Behavior

The cyclic voltammograms of the materials were evaluated in two electrolytes. In 0.5 M H$_2$SO$_4$, the commercial AC and AC-600-800 did not show a significant capacitive behavior (see Figure 10A), probably because the oxygen groups within the material are protonated. The curve observed with the commercial AC has an almost rectangular shape, which is characteristic for electrodes in EDLCs when a charge is stored by only adsorption-desorption mechanisms [43]. The AC-RuO$_2$-40 electrode shows an almost rectangular shape, but there are evident oxidation and reduction signals around 0.4 V. The anodic current at 0.4 V was plotted against the scan rate (Figure 10B), and a linear behavior was obtained. It indicates that there is a high pseudocapacitance component where the charge storage is influenced by the oxidation and reduction of Ru in H$_2$SO$_4$ solution by electrochemical protonation [6].

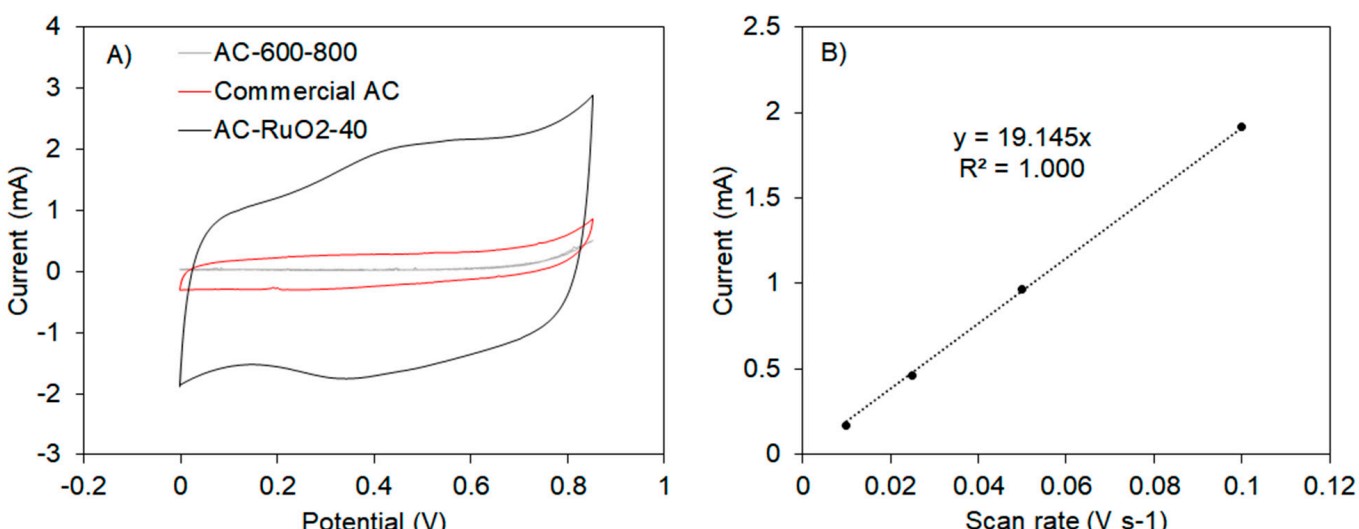

**Figure 10.** (**A**) CV of electrodes made of primed current collector and three different active materials. Electrolyte: 0.5 M H$_2$SO$_4$, Scan rate: 100 mV s$^{-1}$. (**B**) Behavior of the current obtained at 0.4 V during the CV of AC-RuO$_2$-40 at different scan rates. AC-600-800.

The capacitance of the materials with higher current in the cyclic voltammogram, that is, commercial AC and AC-RuO$_2$-40%, was calculated with the integration of the cyclic voltammograms using the following equation:

$$C = \frac{1}{2\,m\,v\,\Delta V} \int_{V_i}^{V_f} I\,dV \tag{3}$$

where $m$ is the mass of the active material loaded on the electrode, v is the scan rate, I is the current, and $V$ is voltage. The capacitance values obtained were 0.268 and 1.62 Fg$^{-1}$ for commercial AC and AC-RuO$_2$-40%, respectively. These low values can be related with loss of the material during the preparation of the electrode, and it deserves more research.

Figure 11A shows the cyclic voltammogram of the AC-600-800 material in 6 M KOH. The CV shows an almost rectangular shape whithin a large potential window ($-1.0$–$0.0$ V), and also higher currents than those obtained with sulfuric acid. It indicates that the high stability of the material and the capacitance behavior due to is high surface area, that appears in basic media. The galvanostatic charge-discharge curve (see Figure 11B) shows a triangular shape and the capacitance obtained with the slope gives a specific capacitance of 393 Fg$^{-1}$.

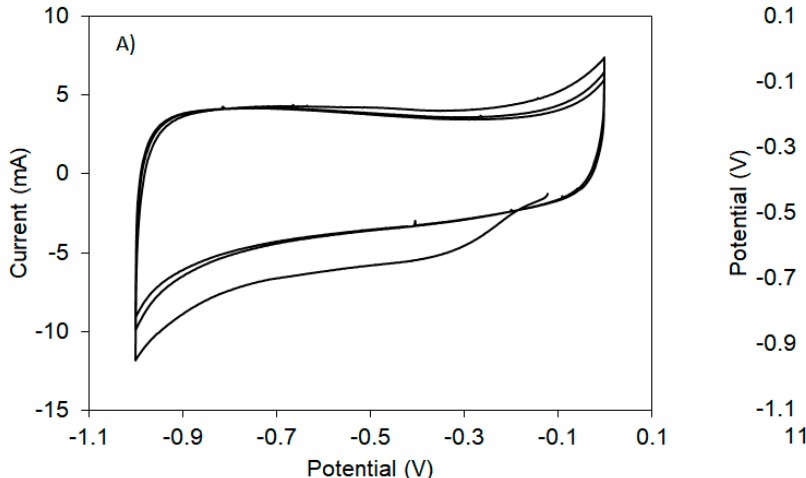 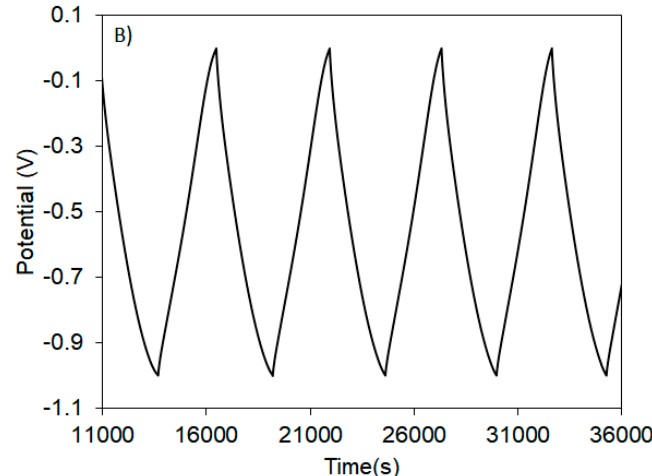

**Figure 11.** (**A**) CV of AC-600-800 in 6 M KOH electrolyte, scan rate: 1 mV s$^{-1}$. (**B**) Galvanostatic charge-discharge curves using AC-600-800 in 6 M KOH with specific current of 0.15 A g$^{-1}$.

The materials with RuO$_2$ were also evaluated with cyclic voltammetry (see Figure 12). For AC-RuO$_2$-10, the CV shows peaks at $-10.8$ and $-10.34$ V vs. Hg/HgO at a scan rate of 5 mV s$^{-1}$. For AC-RuO$_2$-40, clear redox peaks are observed at $-10.91$ and $-10.87$ V under the same conditions as for AC-RuO$_2$-10. The resulting cyclic voltamograms show that even after 30 cycles, there is still redoxcapacity, which proves the electrochemical stability of the materials. The inset also shows an almost rectangular shape between $-10.70$ and $0.0$ V. The specific discharge capacity was calculated by dividing the absolute discharge capacity through the active material mass loaded on the electrodes ($m_{AM} = 0.01$ g). For AC-RuO$_2$-10, a specific discharge capacity of 82 mAh g$^{-1}$ was obtained, whereas AC-RuO$_2$-40 showed a higher specific discharge capacity of even 101 mAh g$^{-1}$. In this case, charge–discharge curves also show a triangular behavior. (see Figure 12B) The specific capacitances obtained for AC-RuO$_2$-10 and AC-RuO$_2$-40 were 336 and 360 Fg$^{-1}$.

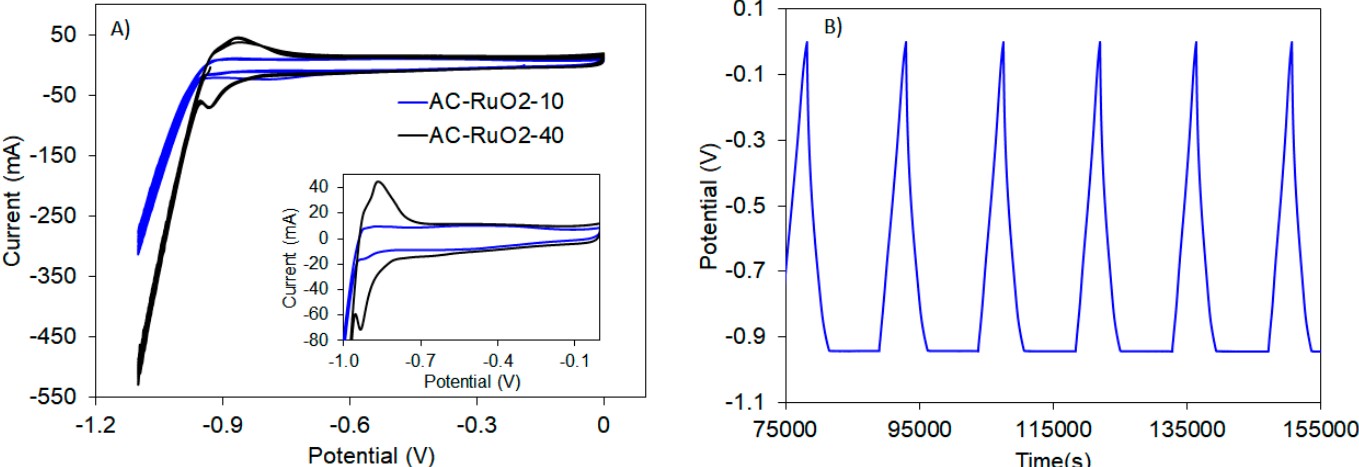

**Figure 12.** (**A**) CV of AC-RuO$_2$-10 and AC-RuO$_2$-40 in 6M KOH electrolyte, scan rate: 5 mV s$^{-1}$. Inset: n = 30. (**B**) Galvanostatic charge-discharge curves using AC-RuO$_2$-10 in 6 M KOH with specific current of 0.10 A g$^{-1}$.

These results show an important relation between the surface area of the materials, their conductivity, and their capacitance. RuO$_2$ based materials have lower surface area than AC-600-800, but higher conductivity and almost the same capacitance. As demonstrated before, the surface area diminishes with RuO$_2$ because the crystallites block the pores of the activated carbon. However, RuO$_2$ high crystallinity (see XRD patterns in Section 3.1.5) allows a better electric conductivity, which favors the electronic transfer in the pseudocapacitor. Moreover, a lower resistance of the material avoids energy losses due to the ohmic drop.

Furthermore, it has to be considered that additives also had an influence on the capacitive behavior. In general, the total amount of additives should be as low as possible, particularly regarding future industrial applications. Since any increase in mass means normally higher prices, the quantity of additives in industry moves in a range of 1–5 wt.% for AB and 2–8 wt.% for binder [44]. As higher binder contents bring a certain risk of clogging the pores of the active material [11], here, it was decided to choose the lowest binder content possible (5 wt.%).

Compared to transition metal oxide-carbon composites based on commercial AC and RuO$_2$ [45], the studied bio-based AC shows very good properties with respect to specific surface area, electric conductivity, and electrochemical performance. Ma et al. [45] also reported a decrease in specific surface area after doping the AC with RuO$_2$ but at the same time measured a specific capacitance almost twice as high as for the pure AC electrode. The resulting decrease in specific surface area corresponds to what was observed for the bio-based carbon composites obtained from corncobs. However, the latter showed specific capacitances one order of magnitude higher than those obtained in [45], which is mainly ascribed to the contribution of the higher specific surface area of the non-doped AC and the higher content of RuO$_2$, which gives rise to an increased pseudocapacitance due to faradaic redox reactions.

## 4. Conclusions

By thermal treatment of corncobs and impregnation of the activated carbon with RuO$_2$, composite carbon materials with good electric conductivity and higher capacitance than commercial activated carbons were produced. Although the specific surface area of the materials decreased significantly after the impregnation with RuO$_2$, the obtained activated carbon–metal-oxide composites showed high chemical stability and better electrochemical performance in terms of electric conductivity. The activated carbon and the materials with RuO$_2$ showed specific capacitances around 350 Fg$^{-1}$. The activated carbon from corncobs, which was used as a precursor for the impregnation step, already had a large specific

surface area of 3.145 m$^2$ g$^{-1}$ with a high content (87%) of micropores. This structure in combination with the high C content of >95 wt.% is assumed to be the main reason for the good physicochemical and electrochemical properties of the final composite material. Considering the fact that corncobs and other lignin-rich waste biomass represent an economic and abundant resource, the production of bio-based carbon composite materials for application in high-performance supercapacitors is a sustainable and promising recycling path for waste biomass towards environmentally friendly energy storage systems.

**Author Contributions:** Conceptualization, V.H., C.R.C., and S.S.; methodology, V.H., C.R.C., S.S., A.d.P.S.-R., M.Q., A.B.B., M.Z., J.P.R.E., M.T.C., J.M.C.M.P., M.-M.T., and A.K.; software, V.H., C.R.C., S.S., A.d.P.S.-R., M.Q., A.B.B., M.Z., J.P.R.E., M.T.C., J.M.C.M.P., M.-M.T., and A.K.; validation, V.H., C.R.C., and S.S.; formal analysis, V.H.; investigation, V.H.; resources, V.H.; data curation, V.H.; writing—original draft preparation, V.H., C.R.C., S.S., A.d.P.S.-R., M.Q., A.B.B., M.Z., J.P.R.E., M.T.C., J.M.C.M.P., M.-M.T., and A.K.; writing—review and editing, V.H.; visualization, V.H.; supervision, M.T.C. and A.K.; project administration, V.H.; funding acquisition, V.H. All authors have read and agreed to the published version of the manuscript.

**Funding:** Financial support by the Ministry of Science, Research and the Arts of Baden-Württemberg and the graduate program BBW ForWerts by means of a LGF grant for V. Hoffmann is gratefully acknowledged.

**Data Availability Statement:** The data presented in this study are available on request from the corresponding author. The data are not publicly available due to institution guidelines.

**Acknowledgments:** Support by the Institute for Manufacturing Technologies of Ceramic Components and Composites (IMTCCC) from University of Stuttgart by providing zeta potential measurements and by the Center for Solar Energy and Hydrogen Research Baden-Württemberg (ZSW) in Ulm for providing electrochemical analysis in KOH electrolyte is gratefully acknowledged.

**Conflicts of Interest:** The authors declare no conflict of interest. The funders had no role in the design of the study; in the collection, analyses, or interpretation of data; in the writing of the manuscript, or in the decision to publish the results.

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
