# Peer review of "Activated Carbon from Corncobs Doped with RuO2 as Biobased Electrode Material"

_electronicmat, doi:10.3390/electronicmat2030023_

Round 1
Reviewer 1 Report
In this manuscript by Hoffmann, the bio-based activated carbon was prepared and it was combined with RuO2 for electrode of capacitor. The material is well characterzied. However, the electrochemical performances of electrodes are not well tested. Thus, before it can be published, the following issues should be well concerned in the revision.
- The mass production of AC to corn cob should be mentioned in the revised manuscript.
- FTIR spectra for AC are necessary to show the groups on the surface.
- The actual mass contents of RuO2 in the composites of AC-RuO2-10 and B) AC-RuO2-40 can be determined by TG analysis in air. In the manuscript, the calculated content is not solid.
- The patterns in figure 6a and 6b can be merged in a picture.
- In addition to CV curves, the charge-discharge curves of the electrodes must be shown in the revision.
Author Response
Dear Sir or Madam,
We appreciate the interest that the reviewers have taken in our manuscript and the constructive criticism they have given. We have addressed the major concerns of the reviewers and we have included a point-by-point response to the reviewers in addition to the changes made in the manuscript. These changes have clearly improved our manuscript.
Thank you.
Sincerely,
Viola Hoffmann

Reviewer 2 Report
Dear Authors
In the present work, the production of Bio-based activated carbons with very high specific surface area of > 3.000 m² g-1 and a high proportion of micropores is described. This activated carbon is doped with different proportions of the highly pseudocapacitive transition metal oxide RuO2 to obtain enhanced electrochemical properties and tune the materials for the application in electrochemical double-layer capacitors. The activated carbon and composites are extensively studied regarding their physico-chemical and electrochemical properties.
The article represents a practical approach to development of carbon based electronics. It is well written and understandable. The quality of figures is good enough. The topic is relatively novel and will be of interest to engineers and material scientists dealing with carbon based electronics.
In that case, I would recommend to be published as it is.
Author Response
Dear Sir or Madam,
we appreciate the interest that the editors and reviewers have taken in our manuscript and the constructive criticism they have given.
Thank you very much for the positive feedback on our manuscript.
Sincerely,
Viola Hoffmann
Reviewer 3 Report
Following points should be considered.
The authors should show that the specific surface area was obtained from the CO2 adsorption experiment in the abstract.
In the experimental section, the range of relative pressure for the calculation of BET analysis should clearly be shown.
Because of the difference in the density, it is quite natural that RuO2/activated carbon composite showed a relatively small specific surface area. RuO2 is assumed to dense particles? What is the actual size of RuO2 particles??
The discussion of XRD pattern is insufficient. Peaks assigned with green lines are carbon? What kind of carbon? Why the intensity of the peaks of carbon increased in the sample of AC-RuO2-40? The intensity of the XRD peaks is not quantitative. The increase in the intensity is also caused by the increase in the size of crystallites. This point should be discussed.
Assignment of the peak at around 38° should be shown.
The electronic conductivity of the materials with binder and conductive additives is mentioned in the discussion of Figure 9. What is the meaning of these values? This value would be necessary for the EDLC capacitors, but the reviewer cannot understand why the authors only mentioned the value in the discussion of Figure 9.
On the other hand, the conductivity shown in Figure 9 is not so important for the evaluation of EDLC. Why the authors reported these results??
What is the advantage of the EDLC using the present AC? Results of EDLC with RuO2-commercial AC composite should be reported to discuss the effect of AC with large surface area in the present study.
Author Response
Dear Sir or Madam,
we appreciate the interest that the reviewers have taken in our manuscript and the constructive criticism they have given. We have addressed the major concerns of the reviewers and we have included a point-by-point response to the reviewers in addition to the changes made in the manuscript. These changes have clearly improved our manuscript.
Thank you.
Sincerely,
Viola Hoffmann

Round 2
Reviewer 1 Report
After revision, I can suggest its acceptance for publication.
Author Response
Dear Sir or Madam,
we appreciate the interest that the reviewers have taken in our manuscript and the constructive criticism they have given.
Thank you for your positive comment.
Sincerely
Viola Hoffmann
Reviewer 3 Report
In Figure 6, the standard XRD pattern for RuO2 (blue lines) does not have the peak at around 38o. Reference for standard pattern (JCPDS # or ICSD#) should be reported. Based on a reference, the assignment of the peak at 38o should be discussed.
Author Response
Dear Sir or Madam,
we appreciate the interest that the reviewers have taken in our manuscript and the constructive criticism they have given. Thank you again for your comment. We adapted the manuscript according to your recommendation.
We extended the discussion with respect to the XRD pattern of RuO2 and discussed the mentioned peak based on additional references (please see line 420-426).
These changes have clearly improved our manuscript. Thank you.
Sincerely
Viola Hoffmann
Round 3
Reviewer 3 Report
The simulated XRD pattern of RuO2 shown in blue lines in Fig.6 is based on rutile structure RuO2 (e.g. JCPDS #43-1027) (This point should be described in the manuscript).
As added by the authors this revision, the peak at 38o could be assigned to RuO2 of JCPDS#50-1428, different crystalline phase, but there are some possibility of metallic Ru. This is the reason why the reviewer ask the assignment of the peak at 38o again.
Author Response
Dear reviewer,
Thank you again for your interest and the comments regarding the discussion of the XRD patterns in our manuscript. We adapted the discussion again based on JCPDS#50-1428 and JCPDS#06-0663.
Best regards
Viola Hoffmann